# Effects of Mesenchymal Stem Cell Injection into Healed Myocardial Infarction Scar Border Zone on the Risk of Ventricular Tachycardia

**DOI:** 10.3390/biomedicines11082141

**Published:** 2023-07-29

**Authors:** Eun-Hye Park, Jin-Moo Kim, EunHwa Seong, Eunmi Lee, Kiyuk Chang, Young Choi

**Affiliations:** 1Cardiovascular Research Institute for Intractable Disease, College of Medicine, The Catholic University of Korea, Seoul 06591, Republic of Korea; park1119@catholic.ac.kr (E.-H.P.); amyeonsohon@nate.com (J.-M.K.); seongeh84@gmail.com (E.S.); dmsaltks@naver.com (E.L.); kiyuk@catholic.ac.kr (K.C.); 2Division of Cardiology, Department of Internal Medicine, Seoul St. Mary’s Hospital, College of Medicine, The Catholic University of Korea, Seoul 06591, Republic of Korea

**Keywords:** mesenchymal stem cells, ventricular tachycardia, myocardial infarction, connexin 43, ambulatory electrocardiographic monitoring

## Abstract

The scar border zone is a main source of reentry responsible for ischemic ventricular tachycardia (VT). We evaluated the effects of mesenchymal stem cell (MSC) injection into the scar border zone on arrhythmic risks in a post-myocardial infarction (MI) animal model. Rabbit MI models were generated by left descending coronary artery ligation. Surviving rabbits after 4 weeks underwent left thoracotomy and autologous MSCs or phosphate-buffered saline (PBS) was administered to scar border zones in two rabbits in each group. Another rabbit without MI underwent a sham procedure (control). An implantable loop recorder (ILR) was implanted in the left chest wall in all animals. Four weeks after cell injections, ventricular fibrillation was induced in 1/2 rabbit in the PBS group by electrophysiologic study, and no ventricular arrhythmia was induced in the MSC group or control. Spontaneous VT was not detected during ILR analysis in any animal for 4 weeks. Histologic examination showed restoration of connexin 43 (Cx43) expression in the MSC group, which was higher than in the PBS group and comparable to the control. In conclusion, MSC injections into the MI scar border zone did not increase the risk of VT and were associated with favorable Cx43 expression and arrangement.

## 1. Introduction

Acute myocardial infarction (MI) produces ischemic scars in the ventricle. Even after recovery from the acute phase of MI, fatal ventricular tachycardia (VT) can occur from the abnormal electrical tissue in the ischemic scar that results in sudden cardiac death [1]. For the prevention of sudden cardiac death associated with VT, an implantation of cardioverter defibrillator is recommended in selected patients after MI, however, this is not a sufficient treatment in patients with recurrent VTs [2]. Ischemic VT is caused by slow conduction areas in the border zone of ischemic scars, where fibrosis and abnormal repolarization create a substrate for reentry [3]. Achieving a conduction block at the slow conduction zone located in a critical isthmus of an ischemic scar terminated the VT in human and animal studies [1,4]. Currently, ischemic VTs are treated with catheter-based radiofrequency ablation, however, the overall success rate is not yet satisfactory, with an acute complication rate >10% [5,6,7]. 

Stem cell transplantation is a promising therapeutic approach for ischemic heart disease [8]. MSC injections into the ventricular myocardium after MI have resulted in reverse-remodeling of the myocardium and improvement in ventricular function [9], however, its effect on ischemic ventricular arrhythmia has not been clearly demonstrated. The scar border zone is the main environment susceptible for post-MI VT, and the direct injection of stem cells into this area would be more effective for improving electrophysiologic properties than the stem cell implantation in infarcted myocardium with a non-viable scar [10]. However, results of experimental studies indicated that MSC therapy could be proarrhythmic due to altered electrophysiologic properties that can result in slowed conduction velocity and the changes in substrate refractoriness in MSC-survived heart tissue [11]. In the present study, the effects on arrhythmogenic risks of MSC implantation into a selective area of the scar border zone in a healed MI animal model were investigated. 

## 2. Materials and Methods

All animal experimental procedures were performed in accordance with guidelines and policies approved by the Institutional Animal Care and Use Committee of the Catholic University of Korea (CUMC-2020-0305-03).

### 2.1. Creation of a Rabbit MI Model

The present study utilized male New Zealand white rabbits (*n* = 7) that were 3 months old and weighed between 3.0 and 3.5 kg. These rabbits were maintained under standard conditions in individual cages. The rabbits underwent left thoracotomy under sedation using intramuscular injections of tiletamine and zolazepam (15 mg/kg) combined with xylazine (5 mg/kg). Tracheotomy with mechanical ventilation was performed in all animals. Pericardium was opened to expose the heart and MI was induced by ligating the proximal portion of the left anterior descending (LAD) coronary artery. Successful MI was confirmed based on discoloration of the left ventricular myocardium supported by the LAD coronary artery. The wound was closed after MI induction, and subcutaneous injections of ketoprofen (3 mg/kg) and gentamycin (4 mg/kg) were performed before and for 3 days after the procedure. A sham procedure without LAD artery ligation was performed in one rabbit; the animal was included in the analysis as a sham control.

### 2.2. Rabbit MSC Isolation and Culture

Bone marrow was collected from the rabbits included in the study procedures. The single-cell suspension was obtained by flushing the bone marrow with Dulbecco’s modified eagle medium (DMEM) using a 22-G needle. After RBC lysis, MSCs were cultured in DMEM containing 10% FBS, 1% penicillin-streptomycin at 37 °C, and 5% CO_2_. The medium was replaced twice a week and MSCs were incubated until 80–90% confluency. Then, cells were subcultured using 0.125% trypsin-EDTA solution. MSCs between passages 2 and 3 were used for transplantation. 

### 2.3. Identification and Differentiation of MSCs

The isolated MSCs were labeled with cell surface markers and then assessed using flow cytometry. Fluorescence-labeled CD45 (eBioscience 11-0451, Invitrogen, Waltham, MA, USA), CD34 (eBioscience 12-0341, Invitrogen, Waltham, MA, USA), and CD31 (Sc-13537, Santa Cruz Biotechnology, Dallas, TX, USA) antibodies were used for staining. Cells were diluted in FACS buffer and incubated with antibody for 30 min at 4 °C. After washing with FACS buffer, stained cells in fresh FACS buffer were analyzed using FacsCantoII (BD Biosciences, Franklin Lakes, NJ, USA).

To identify the expression of CD44 and CD29, real-time sequencing was performed. Total RNA was extracted from isolated MSCs using a RNeasy mini kit (Qiagen, Valencia, CA, USA). Primer sequences used are shown below:
GAPDHFaacatcatccctgcctctactg
RctccgacgcctgcttcacCD44Facaccacggatttctgaccac
RactgctgccacttctctctacatCD29Fcgagtacaccatgagccactatta
Rgatgacattgctggagtttgc

To induce osteogenic differentiation, cultured MSCs at passage 2 were treated with OM media containing 10% FBS, 50 ug/mL L-ascorbic acid, 10 nM dexamethasone, and 10 mM beta-glycerophosphate. The cells were cultured for 2 weeks, with the medium being replaced twice a week. After 2 weeks, alizarin red S staining was performed to assess osteogenic differentiation. The cells were fixed with 70% chilled ethanol and washed with distilled water (DW). They were then covered with alizarin red S (Millipore #2003999) and incubated at 37 °C for 30 min. Absorbance was measured at 550 nm for quantitative analysis.

### 2.4. Electrophysiologic Study (EPS) and Echocardiography 

Four weeks after the MI was induced, surviving rabbits were divided into 2 groups at a 1:1 ratio: MSC group (*n* = 2) and phosphate-buffered saline (PBS) group (*n* = 2). All animals underwent repeated left thoracotomy and EPS was performed from the epicardium of the left ventricle (LV). The sham control also underwent the procedure. After LV exposure, bipolar LV epicardial electrograms were obtained using a quadripolar catheter and electrophysiologic recording system (EP WorkMate, Abbott Medical, Chicago, IL, USA). The electrogram signals < 30 Hz and >500 Hz were filtered out. Programmed electrical stimulation (PES) was performed to measure the effective refractory period (ERP) and VT induction thresholds. PES was started with 8 constant-state pacing at a cycle length of 280–350 ms with a single extra-stimulus at a coupling interval of 250 ms, decreased by 5 ms in each step (minimal coupling interval 50 ms). After reaching ERP of the ventricle with a single extra-stimulus, burst ventricular pacing was attempted with a minimum interval of 100 ms to induce VT. If a ventricular fibrillation (VF) or VT was induced, external direct current cardioversion was performed. Each pacing protocol was performed twice and the measurements were averaged. Echocardiographic data were obtained in all rabbits at 4 weeks after MI and at 4 weeks after cell injection therapy. All echocardiographic values were measured three times and averaged in each animal.

### 2.5. Cell Injection and ILR Implantation

After the completion of EPS, two rabbits were administered 2 × 10^6^ MSCs divided into 4 injections into grossly apparent scar border zones in the epicardium (MSC group) (Figure 1). The other two rabbits were administered equivalent volumes of PBS into 4 scar border zones (PBS group). The sham control rabbit did not undergo cell injection procedure after EPS. Before the closure of thoracotomy, ILR was implanted (LinQ, Medtronic, Minneapolis, MN, USA) in the subcutaneous layer of the left chest wall in all animals. ILR was programmed to record all tachycardia events when heart rate was >180 beats/min.

### 2.6. Repeated EPS and Histologic Examination

Left thoracotomy and repeated EPS were performed using the same protocol 4 weeks after cell injection. ILR was removed and evaluated. After completion of the above studies, the heart was collected and fixed in 4% paraformaldehyde. The hearts were embedded in paraffin blocks and sliced into 4 μm sections that were then slide-mounted and treated with 0.1% Triton X-100 for 30 min and blocked with 10% normal goat serum. The slides were treated with anti-connexin 43 (Cx43) antibodies (1:100, ab66151, Abcam, Cambridge, UK) overnight at 4 °C, followed by secondary antibody Alexa 594 (1:500, A21442, Invitrogen, Waltham, MA, USA) and incubated for 1 year at room temperature. Cx43 imaging was recorded using confocal laser scanning microscope (LSM700) after 4,6-diamidino-2-phenylindole (DAPI) staining for 5 min. Images of 5–6 fields per animal were captured and measured at magnifications of ×200 and ×400. The areas labeled with Cx43 (in red) in each image were measured using the ZEN program version 2.3 (blue edition) (ZEISS, Oberkochen, Germany). 

### 2.7. Statistical Analysis

Continuous variables are presented as means ± standard deviations and compared using the Mann–Whitney *U* test for two-tailed analysis or the Kruskal–Wallis test for comparison of the three groups. Categorical variables were compared using the chi-square test or Fisher’s exact test and are presented as the frequency with percentage (%). *p*-values < 0.05 were considered statistically significant. All statistical analyses were performed using R version 3.6.2 (R Foundation for Statistical Computing, Vienna, Austria).

## 3. Results

MI was successfully induced by LAD artery ligation in six rabbits. Two rabbits died within two days after the LAD artery ligation procedure. Among the remaining animals, two rabbits were allocated to the MSC group and two rabbits to the PBS group. The cell injection procedures were successfully conducted 4 weeks after MI in all animals without procedural complications. 

### 3.1. Echocardiographic Results

Baseline echocardiographic data were obtained one day before the cell injection procedure. Baseline echocardiographic measurements were not significantly different between the MSC and PBS groups. Compared to the sham control, left ventricular ejection fraction (LVEF) was lower and left ventricular end-systolic volume (LVESV) was higher in both the MSC and PBS groups. Echocardiogram was repeated at 4 weeks after cell injection; LVEF was higher (*p* = 0.012) and LVESV was lower (*p* = 0.016) in the MSC group than in the PBS group (Figure 2).

### 3.2. VT Inducibility 

In the initial EPS, VT was not induced by PES or burst pacing from the ventricle, and ventricular ERP was not significantly different between the MSC and PBS groups. Four weeks after cell injection procedure, repeated EPS was performed, and VF was induced in one animal in the PBS group by burst ventricular pacing at an interval of 100 ms. Monomorphic VT was not induced by burst pacing or PES from the ventricle in any animal. Ventricular ERP on the follow-up EPS was also similar in the three groups. Sudden death did not occur in any animal within 4 weeks after cell injection procedures. All ILRs were removed and evaluated at 4 weeks after cell injections. Based on the ILRS interrogations, spontaneous ventricular arrhythmia did not occur in any animal. 

### 3.3. Histology and Cx43 Expression

Upon gross examination of the excised heart samples 4 weeks after cell injection, the infarct size of the LV was decreased in the MSC group compared with the PBS group (Figure 3A). Masson’s trichrome staining of the LV sections showed the fibrotic area was reduced in the MSC group compared with the PBS group (Figure 3B). The immunolabeling for Cx43 fluorescence staining showed Cx43 expression in the infarct border zone was reduced in the PBS group; however, Cx43 expression was restored in the MSC group (Figure 4). In quantitative analysis, the Cx43 signal intensity was significantly increased in the MSC group compared with the PBS group and a significant difference was not observed between the sham control and the MSC group (*p* = 0.021 for the comparison between the PBS and MSC groups and *p* = 0.260 for the comparison between the control and MSC groups; Figure 5).

## 4. Discussion

In the present study, the effects of epicardial MSC injection into the left ventricular scar border zone of a healed MI rabbit model were investigated. The procedure was successful, and the risk of ventricular arrhythmias was not increased after MSC injection, which was assessed using both EPS and continuous cardiac monitor for 4 weeks. Furthermore, this treatment was associated with a reduced fibrotic area of infarcted LV and a normalized Cx43 expression at the border zone.

Stem cell-based regeneration therapy has been attempted to improve outcomes after MI [12]. The benefit of stem cell implantation has been mostly shown for the restoration of left ventricular function and remodeling when delivered to the infarcted area [13,14]. In previous preclinical studies, the favorable effect of stem cell therapies for infarcted myocardium on electrophysiologic characteristics have also been demonstrated [15,16,17,18]. Ischemic ventricular arrhythmia primarily occurs due to altered electrophysiological properties in the border zone of the infarcted region, in which stem cell regeneration therapy may exhibit more significant therapeutic effects [1,19]. However, stem cell therapy can also generate heterogeneous tissue with varying conduction properties, which poses a risk of proarrhythmic effects [11,20]. In the present study, autologous MSCs were injected specifically into the scar border zone of the LV and its safety in arrhythmic risks was confirmed based on EPS and continuous ECG monitoring. There are important implications for using ILR in these animals; it allows us to identify a fatal VT at post-mortem analysis when an animal experienced sudden death. Also, in cases where sudden death did not occur, ILR can provide information regarding the frequency of non-sustained VTs or nonfatal sustained VTs, which has significant implication for prognosis. In addition to the results from EPS, the present study further reinforced the clinical safety of MSC delivery to the border zone through 4 weeks of continuous cardiac rhythm monitoring using implantable devices [21]. Although VT was not evident in the PBS-treated group either, they exhibited unfavorable LV remodeling with extensive fibrosis that is associated with long-term risk of ventricular arrhythmia [22,23]. In the MSC-treated group, increase in LVEF and decrease in the fibrotic area were observed that could lead to a potential clinical benefit in the long term. However, due to a small number of study subjects, it should be confirmed through further research whether stem cell implantation in the scar border zone yields reproducible results in terms of LV reverse remodeling.

Heterogenous repolarization and slow conduction velocity are key features of a myocardium vulnerable to arrhythmia, and altered fiber orientation and Cx43 disarray have been suggested as important mechanisms of this unfavorable electrical remodeling after MI [3,24]. The lateralization and decreased expression of gap junctions in the border zone after MI are associated with both slowed conduction and action potential duration dispersion and the occurrence of ventricular arrhythmias [25]. Similar to our study, it has previously been shown that stem cell implantation can restore the expression and arrangement of Cx43 [26]. The effect of MSCs on the gap junction would provide preventive and therapeutic options for ventricular arrhythmia after MI. Mills et al. reported that intravenous infusion of MSCs in a rat MI model tended to reduce arrhythmia inducibility through enhanced electrical viability and Cx43 expression [26]. However, local delivery of stem cells via intramyocardial injection has been shown to cause nonmyocardial cell clusters, leading to an increased risk of arrhythmia [27]. Meanwhile, the histologic study results and long-term cardiac rhythm monitoring in the present study showed no proarrhythmic effects or heterogenous tissue formation associated with local injection of MSCs into border zones. 

Stem cell therapies targeting the border zone in healed MI are promising in terms of treating ischemic VT. In the present study, MSCs were injected epicardially, however several additional factors should be considered before application in clinical practice. Various types of cells are used for regeneration therapy and compared with autologous MSCs or embryonic stem cells. Induced pluripotent stem cells (iPSCs) have advantages because they are derived from more easily accessible somatic cells and can differentiate into various clinically relevant cell types [28]. Although knowledge regarding the effects of iPSC injection into the scar border zone is limited, in a recent study, a dual stem cell injection using MSCs and iPSCs could effectively treat ischemic heart disease with enhanced vascular regeneration in MI models [29]. In addition, the delivery of immunomodulatory cells can promote favorable remodeling after MI [30]. Delivery routes for stem cell therapy other than epicardial injection have also been studied. The use of biologically engineered tissue can further enhance stem cell survival and engraftment [31,32]. This approach involves incorporating stem cells into tissue constructs or scaffolds designed to mimic the native tissue environment to facilitate better therapeutic outcomes. Although not yet widely commercialized, catheter-based endocardial injection would be an effective semi-invasive approach for targeting the scar border zone when performed under guidance from a 3D mapping system [33]. Currently, various treatment modalities are available for ischemic ventricular arrhythmia but the overall efficacy of these approaches remains unsatisfactory [1]. Therefore, regenerative cell therapy targeting the scar border zone warrants further investigation.

The present study had several limitations. First, the number of animals used per condition was insufficient to draw assertive conclusions. For the statistical comparison of quantitative variables, we used repeated measurements per animal; thus, caution is needed in interpreting these results. Also, although no animals experienced induced VT or arrhythmic death during the study procedure, the small number of animals in which MSC implantation was performed renders drawing definitive conclusions regarding safety difficult. Second, because VT was not induced in all animals at the baseline EP studies, the therapeutic effects of stem cell therapy could not be evaluated. Third, the conduction properties or ERP heterogeneity of the locally treated tissue was not evaluated in this study because EPS was performed by stimulating the non-infarcted area of the LV. Finally, the retention and organization of MSCs in the local tissue was not assessed. 

## 5. Conclusions

Epicardial injection of MSCs into the scar border area in a healed MI rabbit model was feasible and not associated with elevated risk of ventricular arrhythmia assessed based on EPS and long-term cardiac rhythm monitoring. Histologic examination of the border zone where MSCs were implanted revealed favorable changes in gap junction expression and arrangement. The scar border zone is a critical arrhythmogenic source of ischemic VTs and further studies are warranted to investigate delivery methods and therapeutic effects of regeneration cell therapy targeting this area.

## Figures and Tables

**Figure 1 biomedicines-11-02141-f001:**
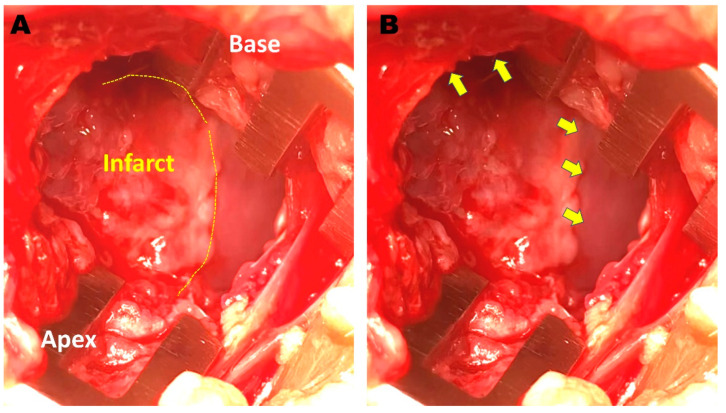
Exposure of rabbit heart after left thoracotomy. (**A**) Infarcted area of left ventricle after left anterior descending coronary artery ligation procedure, and (**B**) scar border zone targeted for cell injections (arrows).

**Figure 2 biomedicines-11-02141-f002:**
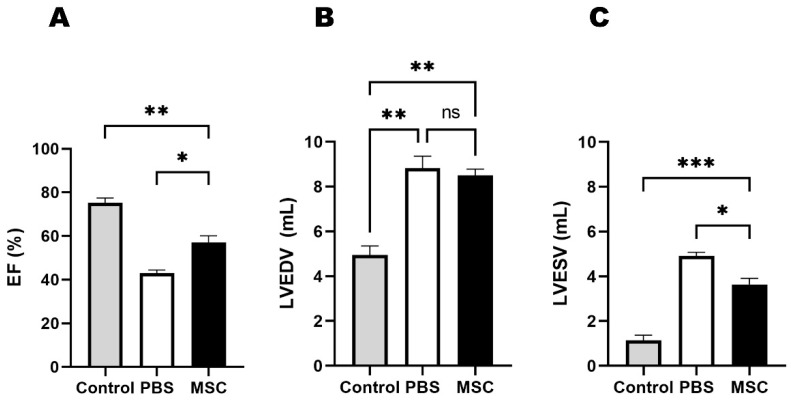
Echocardiographic results in the three groups at follow-up (4 weeks after cell injections). (**A**) Left ventricular EF, (**B**) LVEDV, and (**C**) LVESV. * Indicates *p*-value < 0.05, ** indicates *p*-value < 0.01, *** indicates *p*-value < 0.001 and “ns” indicates no significance. EF = ejection fraction; LVEDV = left ventricular end-diastolic volume; LVESV = left ventricular end-systolic volume; PBS = phosphate-buffered saline; and MSC = mesenchymal stem cell.

**Figure 3 biomedicines-11-02141-f003:**
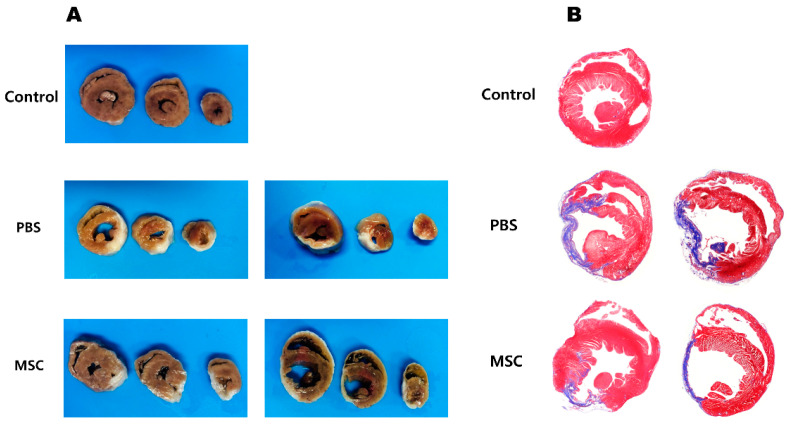
Pathologic examination of left ventricles. (**A**) Gross specimens showing decreased fibrotic area in the MSC group compared with the PBS group. (**B**) Histologic examination after Masson’s trichrome staining shows decreased fibrotic area in the MSC group compared with the PBS group. MSC = mesenchymal stem cell and PBS = phosphate-buffered saline.

**Figure 4 biomedicines-11-02141-f004:**
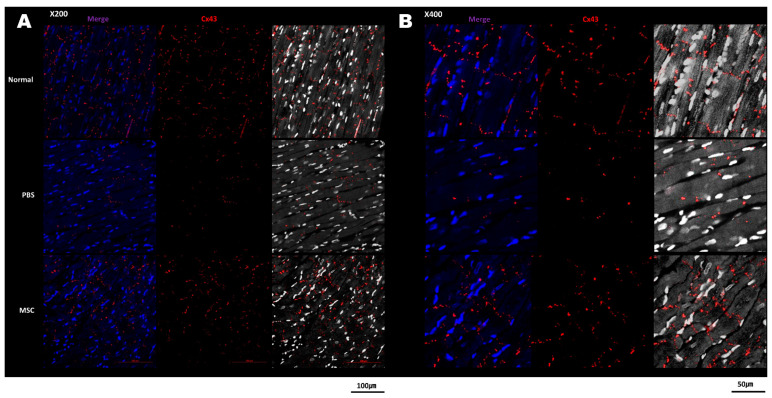
Cx43 expression in the infarct border zone area in the three groups. (**A**) Confocal immunofluorescence imaging shows decreased Cx43 signals (red) in the PBS group and restoration of Cx43 expression in the MSC group. The grayscale image was obtained by deconvolving the maximum intensity DAPI image and merging with Cx43. (**B**) The images at 400× magnification. Cx43 = connexin 43; PBS = phosphate-buffered saline; and MSC = mesenchymal stem cell.

**Figure 5 biomedicines-11-02141-f005:**
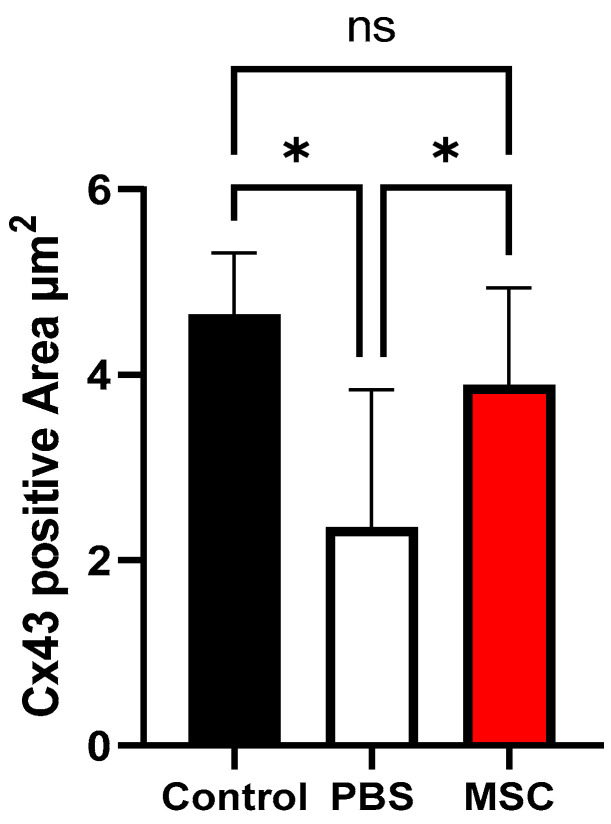
Quantitative analysis for Cx43 expression. Cx43 signal intensity in the infarct border zone was significantly higher in the MSC group than in the PBS group and was similar between the sham control and the MSC group. * Indicates *p*-value < 0.05 and “ns” indicates no significance. Cx43 = connexin 43; PBS = phosphate-buffered saline; and MSC = mesenchymal stem cell.

## Data Availability

The datasets generated and analyzed during this study are not publicly available. Data are available from the corresponding author upon reasonable request.

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
