# Peer review of "Effects of Mesenchymal Stem Cell Injection into Healed Myocardial Infarction Scar Border Zone on the Risk of Ventricular Tachycardia"

_biomedicines, 2023, doi:10.3390/biomedicines11082141_

Round 1

Reviewer 1 Report

The present study addreses the  effects of mesenchymal stem cells (MSC) injection into the border zone of the infarcted left ventricle. The authors report the following findings: i) In a rabbit model of myocardial infarction, MSC injection 4 weeks after MI, improved echocardiographic parameters such as Ejection fraction and Left Ventricular End Systolic Volume. ii) Ventricular effective refractory period was not altered by MSC injection post infarction, suggesting that MSC presence in the border zone of the infarcted heart does not alter the ventricle conduction nor generates cases of spontaneous ventricular arrythmias. iii) A suggested mechanism is indicated by histological results showing that MSC injection increased Connexin 43 expression and attenuated the fibrotic area.

The work of Park et al., is of potential interest due to the ever-growing interest in the community in utilizing stem cell therapy as an attractive therapeutic strategy in several models of disease, such as the discussed possibility of MSC in improving cardiac repair post infarction and its effects on ischemic ventricular tachycardia associated. However, there are several observations that needs to be addressed to strenghten the message of this manuscript.

Major points:

1) The major limitation of this study is the small number of animals used per condition. The starting number of experimental animals (n=7) for 2 conditions is already a limiting factor, which is aggravated by the (predictable) loss of 2 animals due to post-surgery mortality. Of the remaining 5 animals, 2 were used for the MSC injection, 2 as PBS injection control and only 1 animal as a sham control. This issue poses a challenge for the correct interpretation of results which the statistical power is not enough to draw conclusion without overestimating the findings. While statistics are mentioned in the appropiate section, a minimun of n=3 should be considered for correct statistical value that allows for comparison of independent conditions. Moreover, it is not clear how in figure 3, the authors show a dispersion in the sham control condition (First bar) when this would only be represented a single echocardiographic value.

2) The lack of ventricular tachycardia is correctly mentioned as a limitation in the discussion, which effectively demonstrates that injection of MSC does not elevate the risk of ventricular arrythmias post infarction. While this has value in contributing to the discussion of whether MSC slow or not the cardiac conduction, it leaves open the question of how is are MSC improving the cardiac function of the infarcted ventricle and attenuating myocardial fibrosis. While an increased expression and alignment of Connexin 43 would provide benefits in attenuating unfavorable electrical remodeling post infarction, the current model study used by the authors lacks signs of ventricular arrythmias or electrical disturbances that support this interesting possibility. 

Minor points:

3) Figure 6: Higher magnification or resolution of the confocal immunofluorescene images of Connexin 43 could be provided. Also please include scale bars as references for this figure. 

4) Figure 3: Did the authors measure other echocardiographic parameters such as diastolic volumes (LVEV) and wall thickness? While this is a model of infarction with associated systolic dysfunction, it would be interesing to know if the injection of MSC had an associated improvement of diastolic parameters secondary to its effect on the myocardial systolic function.

5) Methodology includes some sections that could benefit by further clarification. These include:

Page 3, line 93: MSCs at passage 2 were incubated in OM Media containing 10% FBS, 50 ug/mL L- 93 ascorbic acid, 10 nM dexamethasone, and 10 mM b-glycophosphate for 2 weeks to investigate the differentiation potency of the isolated MSCs. The medium was replaced twice a week, and after 2 weeks, alizarin red S staining was performedIt is a bit unclear what do the authors mean by investigating the differentiation potency of MSCs. In addition alizarin red S staining use should be better emphasized as a confirmatory staining for MSC identification.

Page 2, line 60: Please indicate the age of the rabbits used

Page 3, line 127: Please indicate how was the immunofluorescence quantification of Cx43 expression (figure 6) performed. How many fields were taken per animal, magnification used, etc..

Reviewer 2 Report

The paper "Effects of mesenchymal stem cell injection into healed myocardial infarction scar boer zone on the risk of ventricular tachykardia" by Park et al is a well-written paper and a good set of data to test the hypothesis, if stem cell injections into the myocardium increase the risk of ventricular tachycardia. They do not have any effects on the incidence of ventricular tachycardias at all.  In contrast, Cx43 expression in rabbit myocardium was increased after stem cell injections. This may implicate, that these stem cells may have been successfully integrated into the myocardial tissue.

Although this study is well-written and thoroughly performed, there a somme minor points that need to be adressed.

1).  Line 70: "The rabbit, where the LAD was not found was used as sham control." This is "intention to treat" and should not be done. The groups of sham and treatment must be divided in advance. Leave this out.  

2.) Fig 2 is not necessary. We all know the site of a loop recorder implantation. If you want to wshow something, sho the ECK, that the loop recorder  generates.

3.) Fig. 3 A and B are not necessary. Just state in the text, that there are no differences in the EF of PBS- and MSC-injected hearts at baseline. That EF ist reduced and LVES is enlarged is clear after LAD ligation.

4.) leave out table 1. The rabbit with ventricular fibrillation in the EP study ist a PBS-control. It is goord enough to state in the text, that cx43 injection in the infarct borderzone does not induce ventricular arrhythmias and does not allow an electrophysiological study to induce ventricular fibrillation.

5.) Leave out figure 4. It is not necessary for the study to show ventricular fibrillation in a control animal.  Above that, the ECG does not show ventricular fibrillation, it does show a ventruclar tachycardia. Just leave it out.

6.) Fig 6 The confocal imager is too dark, one barely can see the coulours. Please make it brighter and add the light microscopical equivalent of this confocal, that you can see the cells. If not possible, leave out and describe in the text.

7.) One Question at the end: Was it necessary to implant loop recoders if nothing ist deteced, except for one control animal ?  If all animals survived, a ventricular fibrillation did not occur. It is really that easy.

Author Response

We very much appreciate the careful review and the very helpful comments of the two Reviewers. We respond to the comments below in normal font and indicate how the manuscript has been modified to respond to these comments. Revisions are indicated in red font in the text.

Reviewer 2

The paper "Effects of mesenchymal stem cell injection into healed myocardial infarction scar boer zone on the risk of ventricular tachykardia" by Park et al is a well-written paper and a good set of data to test the hypothesis, if stem cell injections into the myocardium increase the risk of ventricular tachycardia. They do not have any effects on the incidence of ventricular tachycardias at all.  In contrast, Cx43 expression in rabbit myocardium was increased after stem cell injections. This may implicate, that these stem cells may have been successfully integrated into the myocardial tissue.

Although this study is well-written and thoroughly performed, there a somme minor points that need to be adressed.

1).  Line 70: "The rabbit, where the LAD was not found was used as sham control." This is "intention to treat" and should not be done. The groups of sham and treatment must be divided in advance. Leave this out.  

-> We appreciate your considerate comment. We revised the manuscript as following

Page 2, line 71

“A sham procedure without LAD artery ligation was performed in one rabbit; the animal was included in the analysis as sham control.”

2.) Fig 2 is not necessary. We all know the site of a loop recorder implantation. If you want to wshow something, sho the ECK, that the loop recorder  generates.

->We removed the Fig.2 according to your comment. For EGMs obtained from the ILR, there wasn’t valuable image to provide – all narrow-QRS tachycardias or some noises.

3.) Fig. 3 A and B are not necessary. Just state in the text, that there are no differences in the EF of PBS- and MSC-injected hearts at baseline. That EF ist reduced and LVES is enlarged is clear after LAD ligation.

->The baseline echo data presented as fig 3A.3B were removed. The follow-up echocardiographic measurements were presented in figure 2, and LVEDV graph was added according to the comment from the other reviewer.

4.) leave out table 1. The rabbit with ventricular fibrillation in the EP study ist a PBS-control. It is goord enough to state in the text, that cx43 injection in the infarct borderzone does not induce ventricular arrhythmias and does not allow an electrophysiological study to induce ventricular fibrillation.

-> We removed the table 1 according to your comment.

5.) Leave out figure 4. It is not necessary for the study to show ventricular fibrillation in a control animal.  Above that, the ECG does not show ventricular fibrillation, it does show a ventruclar tachycardia. Just leave it out.

-> We removed the figure 4. Besides, the rhythm was irregular in continuous EGM tracing and demonstrated intermittent fibrillatory firing, which is sufficient to be classified as VF. And it was obviously not a reentrant monomorphic VT which we were trying to induce at the first place, we did not correct the description as VF in the manuscript.

6.) Fig 6 The confocal imager is too dark, one barely can see the coulours. Please make it brighter and add the light microscopical equivalent of this confocal, that you can see the cells. If not possible, leave out and describe in the text.

-> Thank to for the comment. The confocal image was further enhanced for brightness, and an enlarged photograph was attached to improve visibility. The modified image can be found in Figure 4.

7.) One Question at the end: Was it necessary to implant loop recoders if nothing ist deteced, except for one control animal?  If all animals survived, a ventricular fibrillation did not occur. It is really that easy.

-> The question has a valid point. However, there are two main implications for implanting the loop recorder: i) If the animal experienced sudden death, it allows us to determine whether it was caused by ventricular arrhythmia. ii) Even in cases where sudden death did not occur, the presence of the loop recorder enables us to assess the frequency of other ventricular arrhythmias such as non-sustained VT or non-fatal sustained VT, which holds clinical significance. But it was unexpected that we could not find any VT in these MI animals. We have added these findings to the discussion section.

Manuscript change, (Discussion, page 7, line 240)

“There are important implications for using ILR in these animals; it allows to identify a fatal VT at post-mortem analysis when an animal experienced sudden death. Also in cases where sudden death did not occur, ILR can provide information regarding the frequency of non-sustained VTs or nonfatal sustained VTs, which has significant implication for prognosis”

Round 2

Reviewer 1 Report

The authors have made an effort in addresing most of my comments, by strenghtening the discussion, revising the methodology and improving the quality of figures. This is greatly appreciated and helps to deliver the manuscript message in a more clear manner.

My major point about the low number of animals used in the study remains unresolved, and while the authors (understandably) mention that the mortality of animals post infarction was higher than expected, this also shows that the sample number was already on the low limit considering the differental experimental conditions planned in this work (which are 3 at least). While this does not discredit their observations and conclusions, which are correctly proven through their experimental analysis, it does greatly limit the scientific validity and assertive conclusions that one can derive from these observations. 

Author Response

Response to Reviewers’ comments

Reviewer 1

The authors have made an effort in addresing most of my comments, by strenghtening the discussion, revising the methodology and improving the quality of figures. This is greatly appreciated and helps to deliver the manuscript message in a more clear manner.

My major point about the low number of animals used in the study remains unresolved, and while the authors (understandably) mention that the mortality of animals post infarction was higher than expected, this also shows that the sample number was already on the low limit considering the differental experimental conditions planned in this work (which are 3 at least). While this does not discredit their observations and conclusions, which are correctly proven through their experimental analysis, it does greatly limit the scientific validity and assertive conclusions that one can derive from these observations. r

Author’s response: I appreciate your generous comment on the modifications made during the last revision. I also agree with the points you raised. However, the issued of limited number of animals involved in this study is difficult to address at this stage. The validity concerning the effectiveness may be limited in this study. But the primary focus of this research was to assess the feasibility and safety of scar border zone injection rather than to demonstrate treatment efficacy. I believe that the current results hold some significance. In future studies evaluating the therapeutic effects of this treatment, we will ensure validation through a more substantial sample size. Thank you for your valuable comments.